# Fano Resonance in an Asymmetric MIM Waveguide Structure and Its Application in a Refractive Index Nanosensor

**DOI:** 10.3390/s19040791

**Published:** 2019-02-15

**Authors:** Mengmeng Wang, Meng Zhang, Yifei Wang, Ruijuan Zhao, Shubin Yan

**Affiliations:** 1Science and Technology on Electronic Test & Measurement Laboratory, North University of China, No. 3 Xueyuan Road, Taiyuan 030051, China; wmm15534033039@163.com (M.W.); ZM_Bulbasaur@163.com (M.Z.); 18435132117@163.com (Y.W.); ruijuan@nuc.edu.cn (R.Z.); 2School of Instrument and Electronics, North University of China, Taiyuan 030051, China; 3School of Science, North University of China, Taiyuan 030051, China

**Keywords:** plasmonic refractive index sensor, Fano resonance, finite element method, metal–insulator–metal waveguide

## Abstract

Herein, the design for a tunable plasmonic refractive index nanosensor is presented. The sensor is composed of a metal–insulator–metal waveguide with a baffle and a circular split-ring resonator cavity. Analysis of transmission characteristics of the sensor structures was performed using the finite element method, and the influence of the structure parameters on the sensing characteristics of the sensor is studied in detail. The calculation results show that the structure can realize dual Fano resonance, and the structural parameters of the sensor have different effects on Fano resonance. The peak position and the line shape of the resonance can be adjusted by altering the sensitive parameters. The maximum value of structural sensitivity was found to be 1114.3 nm/RIU, with a figure of merit of 55.71. The results indicate that the proposed structure can be applied to optical integrated circuits, particularly in high sensitivity nanosensors.

## 1. Introduction

Surface plasmon polaritons (SPPs) are the electromagnetic wave modes spread along the metal surface [1,2]. They arise from the coupling of incident photons and free electrons on the surface of metals. The electric field intensity of SPPs decays exponentially in the direction perpendicular to the metal-dielectric interface. SPPs break the traditional optical diffraction limit and can realize the transmission and processing of optical information at a nanometer scale [3,4]. Specifically, metal–insulator–metal (MIM) waveguide, which is one of the SPPs-based waveguides and has the merits of strong local field enhancement characteristics, suitable propagation length, and easy integration, and has potential application value in highly integrated photonic circuits [5,6]. There are many optical devices that have been studied, such as plasmonic sensors [7,8,9,10], filters [11,12,13], and splitters [14]. These devices are based on MIM waveguides and made up of waveguides and resonators. 

Special optical effects can be produced by MIM waveguide coupled resonators, such as Fano resonance [15,16,17] and in some cases, electromagnetically induced transparency (EIT) [18,19]. Fano resonance can be considered as an analogy of EIT [20]. Plasmonic systems based on Fano resonance are likely to be highly sensitive sensors because of its sharp and asymmetrical line shape [21,22], owing to which, its transmittance spectrum can be rapidly reduced from the peak to the trough. The full wide half magnitude (FWHM) of this transmission spectrum is relatively narrow and can be easily identified and tracked; thus, greatly improving the sensing resolution [23]. A large number of refractive-index sensor structures based on Fano resonance have been reported [24,25,26,27,28]. This includes a plasmonic Fano system consisting of a stub and a square-cavity resonator with a peak sensitivity of 938 nm/RIU, and the obtained FOM* is approximately 1.56 × 10^5^ [23]. It also includes a symmetric waveguide structure consisting of a baffle coupled with an M-type cavity [29], which can attain a sensitivity of 780 nm/RIU and the FOM* is 1.35 × 10^4^; and a nanosensor consisting of an MIM waveguide with double rectangular cavities, which can achieve a sensitivity of 596 nm/RIU and the FOM is 7.5 [30]. Although the FOM in some works is very high, it is defined differently in each work, and is usually referred to as FOM*. The maximum value of structural sensitivity was found to be 1114.3 nm/RIU, with a figure of merit of 55.71, in this study. It should be noted that the sensitivity is significantly larger than that in other recent works.

In this paper, a novel compact refractive index nanosensor is proposed. The proposed sensor is composed of a baffle coupled with a circular split-ring resonance cavity (CSRRC). Both the transmission spectra and magnetic field distribution of the sensing system are simulated using the COMSOL Multiphysics software based on finite element method (FEM) analysis. The loss of structural symmetry is caused by splitting in the circular resonant cavity, which can excite new resonance modes that cannot be realized by a regular ring. The Fano resonance of the spectrum has varying degrees of dependence on the structural parameters of the system. Hence, the influence of structural parameters, such as the splitting size of CSRRC, the outer radius of CSRRC, and the silver baffle width, on the sensing characteristics is studied.

## 2. Structure Model and Analytical Method

A schematic diagram of the designed waveguide structure is shown in Figure 1. The waveguide consisted of a CSRRC and the MIM waveguide with a baffle. For simplicity, we used a two-dimensional model. To ensure that only the fundamental transverse magnetic (TM_0_) mode was available [31], the width of the MIM waveguide and the CSRRC was fixed at w = 50 nm. The CSRRC was formed by splitting the complete ring. The length of the CSRRC split was represented by *l* and the direction of split was denoted by the angle θ between the center of the split and the x-axis. The inner and outer radii of the CSRRC resonator were *r* and *R*, respectively, and *d* and *g* denote the width of the baffle and the coupling gap size, respectively.

The yellow and white parts in the figure were metallic silver and dielectric air, respectively. The relative dielectric constant of air is $\varepsilon_{d}=1$, and the dielectric constant of silver is obtained using the Debye–Drude dispersion model [32]:(1)$$\varepsilon (\omega )=\varepsilon_{\infty }+\frac{\varepsilon_{s}-\varepsilon_{\infty }}{1+i\tau \omega }+\frac{\sigma }{i\omega \varepsilon_{0}}$$

In Equation (1), the dielectric constant of the infinite frequency is taken as $\varepsilon_{\infty }=3.8344$, the static dielectric constant is taken as $\varepsilon_{s}=-9530.5$, the relaxation time is τ=7.35 × 10^−15^ s, and the conductivity of silver is σ=1.1486×10^7^ S/m.

To analyze the optical response characteristics of the coupled structures, the COMSOL software was used to develop a geometric analysis model. We used ultra-fine meshing to maintain convergence. Certain corresponding parameters, such as the relative dielectric constant of silver, the refractive index of air, and the top and bottom boundaries of the structure were each set using perfectly matched layer PMLs. By simulation calculations, the transmission spectrum of the coupled system was acquired. The transmittance was expressed by $T={\left(S_{21}\right)}^{2}$, where the S_21_ is the transmission coefficient from the input port to the output port (P_1_ to P_2_).

## 3. Simulations and Results

First, we studied the role of the split in the MIM ring. We calculated the transmission spectrum of the MIM waveguide side-coupled complete ring and the CSRRC, respectively. The structural parameters of CSRRC were set as: *R* = 140 nm, θ = 45°, and *L* = 70 nm. The comparison of transmission spectra in the two cases is shown in Figure 2. It contains only the microring resonator and no silver baffle was added. By performing a comparison with the complete ring, a new resonance mode appeared in the CSRRC transmission spectrum at *λ* = 665 nm.

To better understand the transmission characteristics of the structure, we observe Figure 2, which shows the magnetic field distribution H_Z_ of the side-coupled CSRRC structure at the resonant valleys. As shown, the magnetic field distribution exhibited different symmetries. At *λ* = 960 nm, the field distribution was symmetric about the axis across the split and there were two nodes. For the field distribution at *λ* = 665 nm, there were three nodes, and the field distribution was asymmetric about the axis across the split. Therefore, according to the symmetry of the magnetic field, we could divide the resonance mode of CSRRC into a symmetrical mode and an asymmetrical mode. These two resonant valleys correspond to the symmetric mode of *m* = 1 (*λ* = 960 nm) and the asymmetric mode of *m'* = 1 (*λ* = 665 nm), respectively [33]. We attribute the emerging resonance mode to the symmetry loss of the structure.

As shown in Figure 1, a silver baffle was inserted into the MIM waveguide to form the sensor system with CSRRC. The baffle width was *d* = 20 nm. 

The other structural parameters were set as follows: *R* = 140 nm; θ = 45°; *l* = 70 nm; *g* = 10 nm. The transmission spectra of the entire system are shown in Figure 3. We can see that the system (red solid line) had a sharp asymmetrical linearity at *λ* = 955 nm and *λ* = 660 nm, which are typical Fano resonance phenomenon. This phenomenon is formed by the interaction of a broadband continuous state and a narrowband discrete state. For better illustration of the formation of Fano resonance, the entire structure could be divided into the following two structures: a MIM waveguide with baffle and an individual CSRRC resonator. The transmission profile of an isolated MIM waveguide with a baffle (solid black line) corresponds to the broadband continuous spectrum, which has a negative slope and a low transmittance. The transmission profile of the individual CSRRC resonator (solid blue line) has two narrow transmission dips, at *λ* = 665 nm and *λ* = 960 nm. Hence, we consider this as a narrowband discrete state. The two Fano resonances, at *λ* = 955 nm and *λ* = 660 nm, were aroused by the MIM waveguide with a baffle mode interacting with the symmetric mode of m = 1 and the asymmetric mode of m' = 1 of the CSRRC resonator, respectively. We indicate the first-order (*λ* = 955 nm) Fano resonances as Fano(1,1) mode and the second-order (*λ* = 660 nm) as Fano(2,1) mode [23]. Figure 3 shows their corresponding magnetic field distribution.

We then studied the effect of the orientation angle on the resonance characteristics of the system. Because the structure was symmetric about the *y*-axis, we calculated the transmission spectra of the sensor system at orientation angles of θ = −90°, −45°, 0°, 45°, and 90°, as shown in Figure 4. The other parameters were the same as that of Figure 3. There were two phenomena that can be seen from the picture: First, the position of two Fano resonances were nearly unaffected by the orientation angle. Second, the peak intensities of the two Fano resonances varied with the orientation angle. This was because the transmittance, in the two modes of the isolated CSRRC resonator, changed at different orientation angles, thus affected the transmittance of the entire sensor system. The above results show that the peak intensities of the Fano resonance can be changed by adjusting the orientation angle θ. Compared with the complete ring system, the side-coupled CSRRC sensor system can control the resonance more flexibly. By comparison, the transmittance of the Fano(2,1) mode and the Fano(1,1) mode in the transmission spectrum at an orientation angle of θ = 45° was high, which is convenient for observation. Hence, the structure with an orientation angle of θ = 45° was analyzed in this paper.

The Fano resonance was significantly affected by varying refractive indices of the dielectric because of its sharp and asymmetrical line shape [33]. Such a short wavelength change can generate an ultra-narrow transmission peak, which can significantly increase the sensing resolution. Therefore, we studied the sensing properties of the Fano system. Figure 5a shows the transmission spectrum of the structure at different refractive indices, where the refractive index varied from 1.24 to 1.39 RIU (at intervals of 0.03). The structural parameters of the waveguide were the same as that in Figure 3. It is observed from the figure that the two Fano resonances both exhibited obvious redshifts as the refractive index increased. The wavelength shift of the Fano resonance peak changed ∆*λ* as ∆*n* was increased, as shown in Figure 5b. The figure of merit (FOM) and the sensitivity (S) are two major parameters for assessing sensor performance. They are generally expressed as:(2)$$S=\frac{\Delta \lambda }{\Delta n}$$
(3)$$\mathrm{FOM}=\frac{S}{FWHM}$$

From Figure 5b, the peak shift changed linearly as the refractive index increased, which provides a possibility for the application of a refractive index sensor. By linear fitting, the sensitivity of the Fano(2,1) mode was found to be 623.8 nm/RIU, with an FOM = 20.79. The sensitivity of the Fano(1,1) mode was 923.8 nm/RIU, and FOM = 40.16. Due to the higher sensitivity and FOM of the Fano(1,1) mode, it was studied in more detail.

In general, the transmission characteristics of the MIM waveguide structures were affected by the variation of its structural parameters. In this paper, the Fano resonance was caused by the coupling of the MIM baffle and the CSRRC; therefore, the peak intensity and line shape of the Fano resonance was affected by the geometric parameters of the structure. We further studied the influence of the geometric parameters on the Fano resonance.

First, we fixed the other parameters of the waveguide structure at *d* = 20 nm, *l* = 70 nm, and *g* = 10 nm, and increased the CSRRC outer radius *R* from 120 to 160 nm with an interval of 10 nm to study the influence of a single variable on the Fano resonance. The transmission spectra of the structure at different outer radii were calculated, as shown in Figure 6a. As R increased, both Fano resonance peaks produced obvious redshifts. This indicates that the resonance wavelength of the Fano peak was determined by the narrow band discrete state, and the increase of *R* led to the increase of the resonant wavelength in the narrow band spectrum, which caused a redshift of the Fano resonance. The resonance peak shifts of the Fano(1,1) mode with the change in the refractive index Δ*n* are shown in Figure 6b. When *R* = 160 nm, the maximum sensitivity can reach 1114.3 nm/RIU, with an FOM of 55.71.

In addition, we also studied the effect of the silver baffle width on the transmission spectra. The baffle width *d* was varied from 10 to 25 nm in steps of 5 nm, while keeping the other parameters constant. The transmission spectrum obtained by the simulation is shown in Figure 7a. The following observations can be made from the figure: First, the change in the position of the peak was minimal because the field energy was mostly limited to the CSRRC resonator, this can be seen from Figure 3. Thus, the resonance wavelength primarily depends on the geometric parameters of the CSRRC resonator. Second, the line shape of the spectra changed as the width of the baffle increases. This is because the Fano resonance is formed by the interaction of a broadband continuous state and a narrowband discrete state. As the width of the baffle increases, the broad continuous spectrum produced by the isolated MIM waveguide with the baffle will change. This affects the outline of the Fano resonance. Therefore, when the baffle width was changed from 10 to 25 nm, the outline of the Fano resonance changed from being asymmetrical to nearly symmetrical. Figure 7b shows the resonance peak shifts of the Fano(1,1) mode with the refractive index changes Δ*n*. The obtained sensitivity of the sensor system changed only slightly with the increase of d. Furthermore, FWHM became smaller; therefore, FOM got larger.

The other parameters were fixed at *R* = 140 nm, *d* = 20 nm, *r* = 90 nm, and the split length *l* of the CSRRC was changed from 40 to 80 nm (in intervals of 10 nm) to study the influence of CSRRC split length on the transmission spectrum. The simulation results for different split sizes are shown in Figure 8a. As the split length l increased, the Fano resonance peak had an obvious blue shift and the peak intensities of the resonance did not change significantly. Figure 8b shows the relationship of the resonance peak shift of the Fano(1,1) mode with the change in refractive index Δ*n*. The fitting results show that the sensitivity of the sensing system was as high as 960 nm/RIU at *l* = 40 nm, and the FOM was 48. 

## 4. Conclusions

In this study, a tunable refractive index nanosensor system is proposed and analyzed using FEM analysis. The proposed system consists of an MIM waveguide with a baffle coupled with a CSRRC resonator. Simulation results indicate that the waveguide structure can realize dual Fano resonance. The resonance wavelength depends largely on the geometric parameters of the waveguide. The resonance wavelength is especially susceptible to the structural parameters of CSRRC (i.e., *R* and *l*). Additionally, the line shape of the spectra is affected by the baffle width *d*. The peak intensities of the Fano resonance can be changed flexibly by adjusting the orientation angle θ of CSRRC. The designed sensor system has excellent performance. By optimizing the geometric parameters, the obtained sensitivity can reach 1114.3 nm/RIU with an FOM of 55.71. Combining the MIM waveguide based on SPPs yields the advantages of simple and easy integration. These results can provide a new reference for on-chip plasmonic refractive index nanosensors.

## Figures and Tables

**Figure 1 sensors-19-00791-f001:**
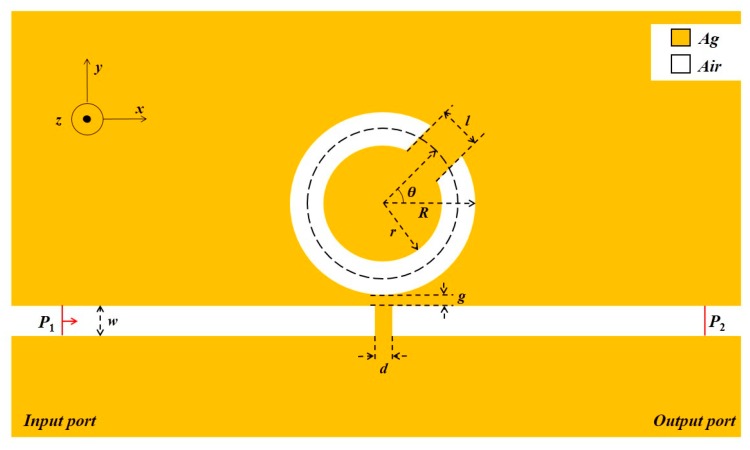
2D Schematic of a plasmonic refractive index nanosensor.

**Figure 2 sensors-19-00791-f002:**
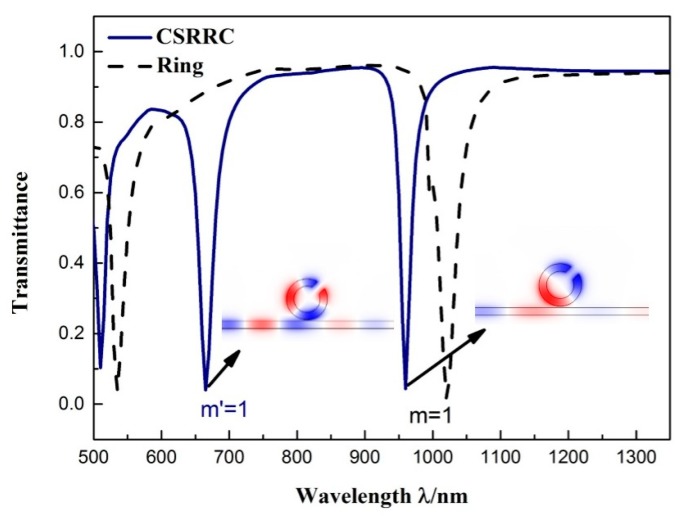
Transmission spectrum of the MIM waveguide side-coupled complete ring and side-coupled CSRRC resonators.

**Figure 3 sensors-19-00791-f003:**
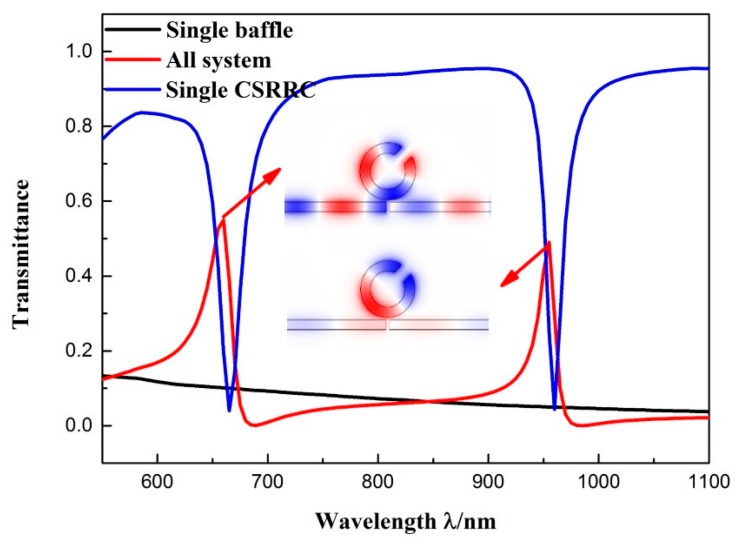
Transmission spectra of individual CSRRC resonator (blue line), MIM waveguide with baffle (black line), and sensor system (red line). The H_Z_ field distribution is shown at the resonant peak.

**Figure 4 sensors-19-00791-f004:**
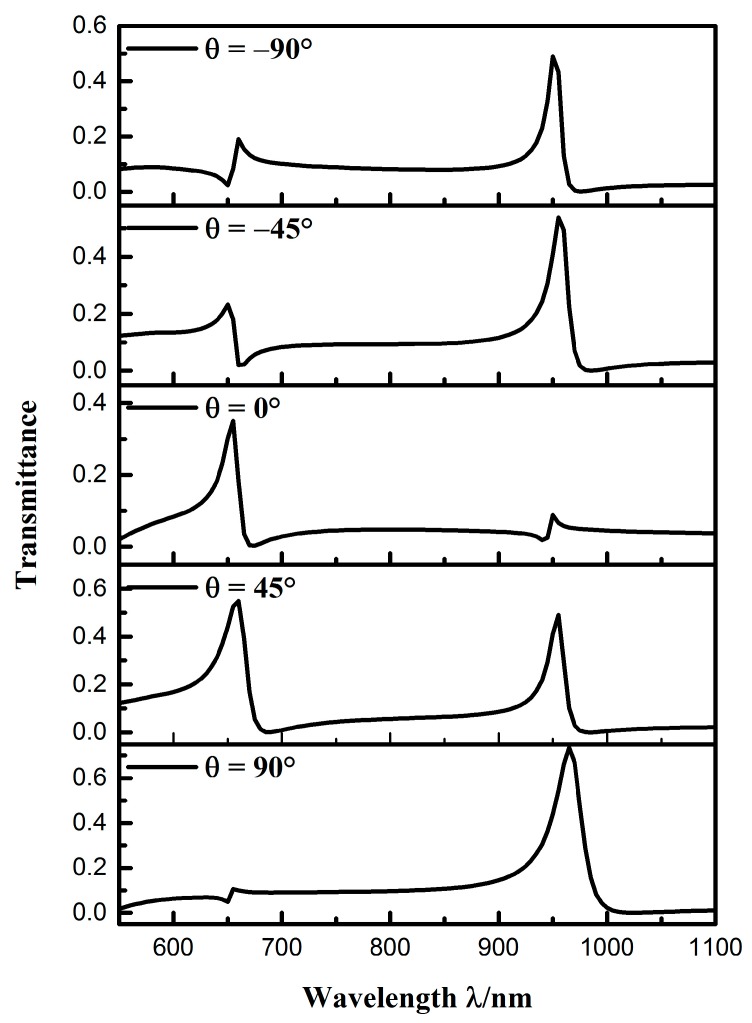
Transmission spectra at different orientation angles.

**Figure 5 sensors-19-00791-f005:**
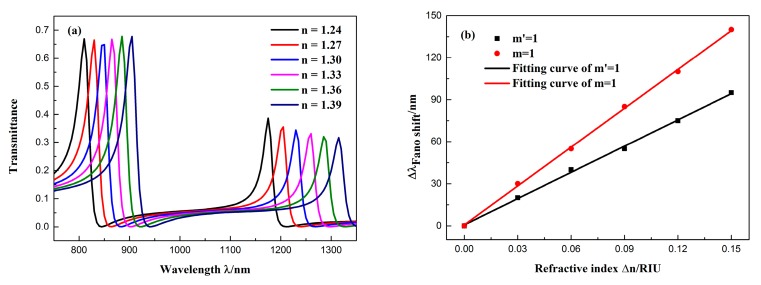
(**a**) Transmission spectra of the sensor at different refractive indices n. (**b**) Fitted line plot of the peak shift (∆*λ*) changes with the change in the refractive index (∆*n*).

**Figure 6 sensors-19-00791-f006:**
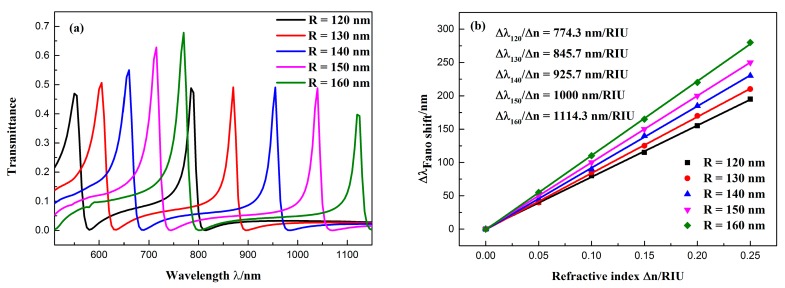
(**a**) Transmission spectra of the sensor at different CSRRC outer radii; (**b**) fitted line plot of the Fano(1,1) mode resonance peak shift (∆*λ*), changing with the increase of the refractive index (∆*n*).

**Figure 7 sensors-19-00791-f007:**
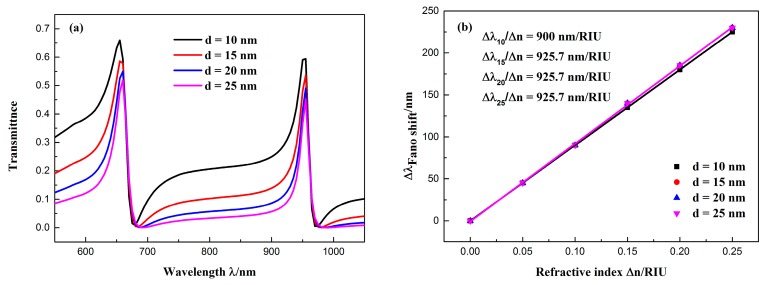
(**a**) Transmission spectra of the sensor at different baffle widths; (**b**) fitted line plot of the Fano(1,1) mode resonance peak shift (∆*λ*) changing as the change in refractive index (∆*n*) increases.

**Figure 8 sensors-19-00791-f008:**
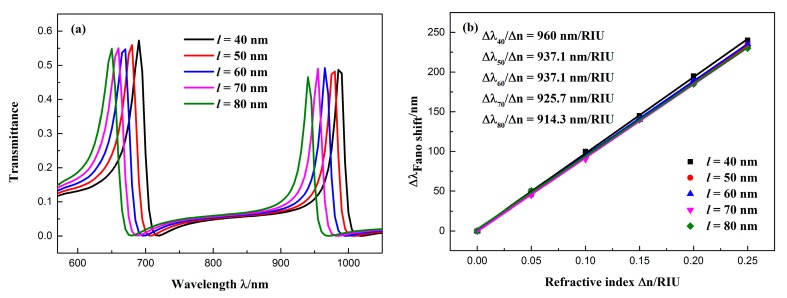
(**a**) Transmission spectra of the sensor at different CSRRC split lengths; (**b**) fitted line plot of the Fano(1,1) mode resonance peak shift (∆*λ*) changes with the increase in the refractive index (∆*n*).

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
