# Peer review of "Fano Resonance in an Asymmetric MIM Waveguide Structure and Its Application in a Refractive Index Nanosensor"

_sensors, 2019, doi:10.3390/s19040791_

Round 1
Reviewer 1 Report
The authors report the controllable Fano resonance in Asymmetric MIM waveguide structure and the refractive index sensitivity.  I think the refractive index sensitivity looks large enough for a report but the description of the position in the manuscript is not enough. Therefore, the manuscript will be suitable for Sensors after the appropriate modification of the manuscript.
1.    Background of the research and the motivation
Many asymmetric MIM waveguide structures with Fano resonances for refractive index sensor have been reported. The author reported the sensitivity and FOM of the refractive index in the structure. Therefore, the authors should state the position of their research relative to the other reports and compare the results here to the other structures. Otherwise, the significance of this manuscript cannot be understood.
2.    Reference format and citation
There are so many errors (disunion) in the description format.  Did the authors really check them?  The authors should check carefully in all references.
1)    Journal name should be abbreviated. Some references are shown in full name.
2)    The issue number should be shown or not?
3)    The page numbers should be shown “from” and “to”.
4)    Reference 24 had been already published.
3.    Main body
1)    [p. 1, L 42] The authors introduced some related reports regarding refractive index sensitivity by the Fano resonance. FOM and FOM* values are important but not shown in the manuscript. The authors should describe them to compare them with the results of the manuscript.
2)    Regarding the Fano resonance in this MIM system, which stats broadband continuous state and narrowband discrete state does correspond to each state in the isolated state?  It is difficult for readers to understand the Fano resonance in this system.
3)    [p. 2, L. 70] What is the purpose did the authors cite the reference [26]? I could not find the dielectric function of Ag in the reference [26].
4)    The variable “l” should be italic or capitalized. There are some variables of “l” in Roman style.
5)    [Figure 3] The magnetic field Hz should be Hz, z should be subscripted.
6)    The authors used “peak values” as the meaning of peak intensity or amplitude.  However, “peak values” contain ambiguity of “peak wavelength” and “peak intensity”. I think “peak intensities” is better than “peak values”.
Author Response
Dear Reviewers:
Thank you for your comments concerning our manuscript entitled “Fano Resonance in Asymmetric MIM Waveguide Structure and its Application in Refractive Index Nanosensor” (Manuscript ID: sensors-438352). Those comments are all valuable and very helpful for revising and improving our paper, as well as the important guiding significance to our researches. We have studied comments carefully and have made correction which we hope meet with approval. Revised portion are marked using the "Track Changes" function in microsoft word of the paper. Besides, we have carefully checked through the whole manuscript and corrected some grammar mistakes. The main corrections in the paper and the responds to the reviewer’s comments are as flowing:Special thanks to you for your good comments.

Reviewer 2 Report
The paper deals with a new theoretical design of a nanophotonic sensor based in Fano resonance in asymmetric MIM Waveguide and its application in refractive index sensor. The design is novel as far as I see but the final performance (sensitivity and FOMs) are average or even lower than other designs. I recommend the authors to review the literature more in detail and compare results with other works, like:
"High-sensitivity refractive index sensors based on Fano resonance in the plasmonic system of splitting ring cavity-coupled MIM waveguide with tooth cavity"
Zhang, Y., Kuang, Y., Zhang, Z. et al. Appl. Phys. A (2019) 125: 13. https://doi.org/10.1007/s00339-018-2283-0
"Tunable Fano resonance in plasmonic MDM waveguide with a square type split-ring resonator"
https://doi.org/10.1016/j.ijleo.2018.06.027
"Optical Fano Resonance in MIM Waveguides with a Double Splits Ring Resonator"
Sensors 2018, 18, 287; doi:10.3390/s18010287
I also recommend to include a discussion clarifying the actual advantages and disadvantages of this design in respect to others, including why should be Fano resonances better than regular resonances in the transmission spectrum of Fig. 2.
Check spelling, there are several typos along the manuscript.
Author Response

(The authors gave the same response as above.)

Reviewer 3 Report
Dear Authors
The work presented in this manuscript is interesting and well organized. However, there is room for some improvements, as described in the following comments:
1) The introduction provides sufficient background and includes a sufficient amount of references. However, there are some recent articles on the development of photonic nanosensors using optical microring resonators which are not mentioned, e.g.:
Optical Ring Resonators: A Platform for Biological Sensing Applications. J Med Signals Sens. 2017;7(3):185-191.
A study on refractive index sensors based on optical micro-ring resonators, Photonic Sens (2017) 7: 217-225. 
2) I suggest that section 3 should be renamed toI  "Simulations and results", because only numerical "experiments" have been performed. 
3) I suggest that it should be made clear that the transmission spectrum in figure 2 corresponds only to the micro-ring resonator, and not to the whole system, because as it is written in the text and in the figure caption it is a bit confusing.  
4) I suggest that more values of the orientation angle [theta] of the gap should be considered. For example, how would the system behave if the gap was located at -45, 135, 180, or 225 degrees?
5) I suggest that the sensitivity of the system on the variations of the refractive index should also be determined in the region of the refractive index of water (around n=1.33), since water is a univeversal solver used in many chemical and biologiacal applications.
In conclusion, I suggest that the manuscript will be suitable for publiction after the above mentioned improvements are made.   
Author Response

(The authors gave the same response as above.)

Round 2
Reviewer 2 Report
I have seen the corrections made and I think the paper can be published.